# Prevalence of *Cryptococcal Antigenemia* and associated factors among HIV/AIDS patients on second-line antiretroviral therapy at two hospitals in Western Oromia, Ethiopia

**Nuguse Geda**[1], **Tafese Beyene**[2], **Regea Dabsu**[1], **Hylemariam Mihiretie Mengist**[ID][3]*

**1** Department of Medical Laboratory Sciences, Institute of Health Sciences, Wollega University, Nekemte, Ethiopia, **2** Department of Medical Laboratory Sciences, College of Health Sciences, Arsi University, Assela, Ethiopia, **3** Department of Medical Laboratory Sciences, College of Health Sciences, Debre Markos University, Debre Markos, Ethiopia

* hylemariam@gmail.com

## Abstract

### Background

Cryptococcosis is a global public health important infectious disease. HIV infection is the main risk factor estimated to account for 95% of cases in the middle- and low-income countries and 80% of the cases in high-income countries.

### Objective

The main aim of the study was to determine the prevalence and associated risk factors of *Cryptococcal antigenemia* (CrAg) among HIV/AIDS Patients on second-line ART Therapy at Ambo General Hospital and Nekemte Referral Hospital, Western Oromia, Ethiopia.

### Materials and methods

Hospital-based cross-sectional study was employed from September 1, 2017, to October 30, 2017. Whole blood was tested for CrAg using Cryptococcal lateral flow assay (Immuno-Mycologics, Norman, OK, USA) according to the manufacturer's instructions. The collected data were analyzed using SPSS version 20 software. Binary logistic regression models were applied to assess the association between predictors and outcome variables at 95% CI.

### Result

Among the study participants, 115(62.8%) were females and the median age of the participants was 35 (IQR: 14) years. Majority, 169(92.3%), have been living with HIV for $\geq$ 5.6 years and 124 (67.8%) stayed on 2nd line ART for an average of 2.5 years. The overall prevalence of *Cryptococcal antigenemia* in the study participants was 7.7% (14/183). Being male [AOR, 95% CI: 4.78(1.14, 20.1)], poor adherence to ART [AOR, 95% CI: 0.12(0.03, 0.4)], occupational exposures to contaminated soil [AOR, 95% CI: 6.81(1.38, 33.4)], having

**Data Availability Statement:** All relevant data are within the manuscript and its Supporting Information files.

**Funding:** The authors received no specific funding for this work.

**Competing interests:** The authors have declared that no competing interests exist.

non-separated house from chickens [AOR, 95% CI: 0.06(0.01, 0.51)], CD4 T+ cell levels ≤ 100 cell/µL [AOR, 95% CI: 6.57(1.9, 23.3)] and viral load >1000 copies/mL [AOR, 95% CI: 11.7(2.4, 57.8)] were significant predictors of *Cryptococcal antigenemia* (P≤ 0.05).

## Conclusion

The prevalence of *Cryptococcal Antigenemia* was significantly high in this study. Being male, occupational exposure to contaminated soil with avian droppings, CD4+ T cell levels <100 cell/µL and viral load >1000 copies/mL were significant predictors of C*ryptococcal* antigenemia. Therefore, public health measures, adherence to ART and early treatment are recommended.

## Introduction

Cryptococcosis is a zoonotic disease and it is one of the life-threatening opportunistic mycoses in HIV/AIDS patients. The disease is caused by a capsulated yeast of the *Cryptococcus neoformans* complex [1]. The yeast is found in the environment mostly associated with pigeon droppings, soil and avian excreta throughout the world [2]. The yeast mainly causes pulmonary infection upon inhalation of the spores and symptoms are pneumonia-like. In immunosuppressed patients, the fungi disseminate to other sites commonly to the meninges and cause life-threatening Cryptococcal meningitis [3].

Cryptococcosis is the commonest fungal disease in immunocompromised individuals and it infects more than one-third (60–70%) of HIV/AIDS patients [4, 5]. The epidemiology of crptococcosis has been increased in the past three decades governed by the progress of the HIV pandemic. Although a combination of treatments (ART and anti-fungal drugs) has been contributing to the dropping of mortality in high income countries (HICs), the death rate in middle and low income countries (MLICs) is still high. Specifically, CNS cryptococcosis is deadly unless treated [4].

Detecting the presence of cryptococcal antigens in the patients' blood, *Cryptococcal antigenemia* (CrAg) tests, is the commonest method for diagnosing *Cryptococcal meningitis* (CM). The CrAg test can detect the target antigen from peripheral blood on average of 22 days prior to the development of CM and it is estimated that about 11% of patients will have positive CrAg test more than 100 days prior to the onset of the disease, CM [5].

The survival rate and the quality of life of HIV/AIDS patients with cryptococcosis have been greatly improved after the administration of a combination of antiretroviral and antifungal therapy [6,7]. However, HIV/AIDS patients with CD4+ T cell counts <100 cells/µL are at risk of Cryptococcosis and the situation is worse in those who are not taking ART. Therefore, early screening and treatment of HIV/AIDS patients for cryptococcosis is of a vital significance [8].

Although previous studies in Ethiopia reported about the prevalence of *cryptococcal antigenemia* in HIV/AIDS patients, there is a paucity of published data regarding *cryptococcosis* among HIV/AIDS patients under second-line ART specifically in the study area. This study is, therefore, aimed to determine the prevalence and associated risk factors of *Cryptococcal antigenemia* among HIV/AIDS patients on 2nd line ART at Ambo General Hospital and Nekemte Referral Hospital of Western Oromia, Ethiopia. We believe this is the first study in the specific study area and will contribute to the betterment of prevention and treatment activities of crptococcosis in Ethiopia

## Materials and methods

### Study setting and context

A hospital-based cross-sectional was conducted at Ambo General Hospital and Nekemte Referral Hospital, Western Oromia, Ethiopia, from September 1, 2017, to October 30, 2017. Ambo and Nekemte towns are located at 115 and 327 kilometers West of the Capital Addis Ababa, respectively. These two study areas were selected by lottery method from eight western Oromia hospitals which serve for a significant number of HIV/AIDS patients on second-line ART. Adult HIV/AIDS patients on second-line ART regimens were 129 and 154 at Ambo General Hospital and Nekemte Referral Hospital, respectively.

**Study population.** The study population for this study was all consented HIV/AIDS patients on $2^{nd}$ line ART at Ambo General Hospital and Nekemte Referral Hospital. All HIV/AIDS patients under second-line anti-retroviral treatment aged $\geq$18-years-old who didn't take anti-cryptococcosis drugs in the past three months were included in the study.

**Sample size determination and sampling technique.** The sample size was calculated using a single population proportion formula taking 4.1% prevalence of *Cryptococcal Antigenemia* from previous study [9], a margin of error 5% and 95% confidence level.

- $n = (Z\alpha/2)^2 \, p \, (1-p)/d^2$

- $Z\alpha/2$ = standard normal variable at 95% confidence level (1.96).

- d = precision (tolerable margin of error)

- $n = Z\alpha/2 \, p \, (1-p)/w2 = (1.96)^2 \times 0.041 \, (1-0.041)/(.05)2 = 60$ patients

- Considering 15% as non-response rate, the final sample size is: $(60 * 0.15) + 60 = 69$ participants.

    By using Design Effect = 2.7

- $n = 69 * 2.7 = 186$ participants.

The study participants were then selected from the two hospitals using proportional cluster sampling technique as follow:

$$n_i = \frac{n}{N} N_i$$

- $n_i$ is a sample size of the $i$hospital

- $N_i$ is a total of HIV-infected patients on second-line ART registered for ART in $i$hospital.

- $n = n_1 + n_2$ is the total sample size, which is 186

- $N = N_1 + N_2$ is the total number of HIV-infected patients on second-line ART registered for ARTin the two hospitals.

Finally, a total of 186 (101 from Nekemte Referral Hospital and 85 from Ambo General Hospital) study participants were consecutively enrolled.

**Data collection.** Site assessment and pretest was done prior to the actual data collection in an unselected hospital using 5% of sample size and data were collected using structured questionnaires. The questionnaire was developed in English and translated into the local language; Afan Oromo. Data were collected by trained nurses and medical laboratory technologists. Collected questionnaires were cross-checked daily for completeness.

**Laboratory procedures.** Whole blood was collected from each study participants. CrAg Lateral Flow Assay (CrAg LFA) and CD4+ T-cell counts were done following standard protocols and manufacturer's instructions [10,11]. All CrAg LFA tests were done in duplicate by two laboratory technologists to minimize bias and avoid errors. Briefly, one drop of the patient specimen is added to the EDTA tube and then the CrAg LFA kit is inserted and incubated for 10 minutes. The test is interpreted "positive" if two lines appear on both the control and test lanes, "negative" if one line appears on only the control line and "invalid" if a line appears only on the test lane.

The number of viral RNA copies (viral load) is determined for every HIV/AIDS patient in Ethiopia using Real-Time Reverse transcription HIV-1 Polymerase Chain reaction (RT-PCR) (Abbott MolecμLar, Inc., Des Plaines, IL) at Ethiopian Public Health Institute and Nekemte Regional Laboratory [12] while CD4+ T cell count was done at the respective health facilities. Therefore, we didn't collect additional blood for the purpose of reverse transcription polymerase chain reaction (RT-PCR) and CD4+ T cell count in our study rather we used laboratory data which were done simultaneously with the CrAg test for the purpose of patient follow-up.

**Data quality assurance.** The questionnaires were carefully designed, pre-coded, and then pretested to minimize errors. Data collectors and data assistants were trained prior to data collection. The collected data were checked daily for completeness and callbacks were arranged for any incomplete data. Experienced laboratory personnel was recruited to perform tests using appropriate laboratory facilities which have a separate area for sample preparation and analysis and access to water of appropriate quality. Adequate blood specimens were collected using carefully labeled dry EDTA tubes. The specimens were kept free of contamination from water and soil. All CrAg, CD4+ T cell count and viral load test results were encoded and reported appropriately. All CrAg tests were run together with positive and negative control samples. All reagents were checked for expiry date and all procedures were done following the manufacturer's guidelines and standard operating procedures.

**Data analysis.** Cleaned and coded data were entered into EPI-data version 3.1 software and exported to SPSS version 20 software for analysis. Descriptive data were presented using frequency statistics. Binary logistic regression models were used for inferential statistics to identify independent predictors of Cryptococcal *antigenemia* at 95% CI.

## Ethics approval considerations

The study was conducted after it was ethically reviewed and approved by the Institutional review board (IRB) of the research directorate of Wollega University. Then a letter informing the respective hospitals was written from Wollega University and permission obtained. All the information obtained from the study participants was coded to maintain confidentiality. All participants were adults to provide informed consent in line with local laws. The IRB approved the use of oral consent documented by a witness after the objectives of the study had been explained. The positive results were timely reported to the clinicians for appropriate intervention.

## Results

### Socio-demographic characteristics of study participants

A total of 183 HIV-infected patients on second line ART were enrolled in the study with a 98% response rate. Majority of the participants were females 115(62.8%) and the median age of the study participants was 35 (IQR: 14) years. About 81.4% of them had monthly income <1500 birr, 132(72.1%) were urban residents and 97(53%) were married (**Table 1**).

**Table 1. Socio-demographic characteristics of HIV-infected patients attending second-line ART at Ambo General Hospital and Nekemte Referral Hospital, Western Oromia, Ethiopia from September 1, 2017, to November 30, 2017.**

| Variables | Frequency | Percentages |
| --- | --- | --- |
| **Age (years)** | | |
| 18–28 | 52 | 28.4 |
| 29–38 | 64 | 35 |
| 39–48 | 46 | 25.1 |
| 49–58 | 16 | 8.7 |
| 59–68 | 3 | 1.6 |
| > 68 | 2 | 1.1 |
| **Sex** | | |
| Female | 115 | 62.8 |
| Male | 68 | 37.2 |
| **Marital Status** | | |
| Single | 33 | 18 |
| Married | 97 | 53 |
| Divorced | 19 | 10.4 |
| Widowed | 29 | 15.8 |
| Separated | 5 | 2.7 |
| **Educational Status** | | |
| Illiterate | 51 | 27.9 |
| Read & write | 12 | 6.6 |
| Primary School | 64 | 35 |
| Secondary School | 43 | 23.5 |
| College & above | 13 | 7.1 |
| **Occupation** | | |
| Employee | 29 | 15.8 |
| Farmers | 37 | 20.2 |
| Merchants | 49 | 26.8 |
| Students | 21 | 11.5 |
| Housewife | 22 | 12 |
| Daily Laborer | 13 | 7.1 |
| Others | 12 | 6.6 |
| **Monthly Income** | | |
| ≤ 1500 birr | 149 | 81.4 |
| >1500 birr | 34 | 18.6 |
| **Residence** | | |
| Rural | 51 | 27.9 |
| Urban | 132 | 72.1 |

## Clinical characteristics of study participants

The mean duration of the patients living with HIV was 8.75 years. The mean duration of first-line ART was 4.5 years and 3.7 years for second-line ART. The mean baseline (before starting treatment) CD4+ T cell level of the participants was 163.2±82 cells/µL. The mean current CD4+ T cells count was 434.4±286.3 cells/µL whereas the median viral load was 10110 HIV RNA copies/ml (**Table 2**).

**Table 2. Clinical characteristics of HIV-infected patients attending Second-Line ART at Ambo General Hospital and Nekemte Referral Hospital; Western Oromia, Ethiopia; from Sep. 1, 2017, to Nov. 30, 2017.**

| Variables | | Frequency | Percentages |
|---|---|---|---|
| Years lived with HIV | 0.5–5.5 | 14 | 7.7 |
| | ≥5.5 | 169 | 92.3 |
| Years on first line ART | <0.5 | 13 | 7.1 |
| | 0.5–5.5 | 107 | 58.5 |
| | ≥5.5 | 63 | 34.4 |
| Years on second line ART | <0.5 | 21 | 11.5 |
| | 0.5–5.5 | 124 | 67.8 |
| | ≥5.5 | 38 | 20.8 |
| Adherence to ART | Poor | 39 | 21.3 |
| | Good | 144 | 78.7 |
| The distance of the hospital | <1 Km | 29 | 15.8 |
| | 1–3 Km | 75 | 41 |
| | >3 Km | 79 | 43.2 |
| Occupational Exposures to blood | Yes | 4 | 2.2 |
| | No | 179 | 97.8 |
| Occupational Exposures to soil | Yes | 85 | 46.4 |
| | No | 98 | 53.6 |
| Domestic animals like hen present | Yes | 82 | 44.8 |
| | No | 101 | 55.2 |
| House separated from the hen | No | 15 | 18.3 |
| | Yes | 67 | 81.7 |
| The patient was | Inpatient | 8 | 4.4 |
| | Outpatient | 175 | 95.6 |
| Fever | Present | 45 | 24.6 |
| | Absent | 138 | 75.4 |
| A headache | Present | 93 | 50.8 |
| | Absent | 90 | 49.2 |
| Skin Lesion | Present | 32 | 17.5 |
| | Absent | 151 | 82.5 |
| A cough | Present | 43 | 23.5 |
| | Absent | 140 | 76.5 |
| Neck Stiffness | Present | 8 | 4.4 |
| | Absent | 175 | 95.6 |
| Weight Loss | Present | 38 | 20.8 |
| | Absent | 145 | 79.2 |
| Night Sweatiness | Present | 35 | 19.1 |
| | Absent | 148 | 80.9 |
| Blurred Vision | Present | 12 | 6.6 |
| | Absent | 171 | 93.4 |
| Baseline CD4+ T counts (cells/μL) | ≤100 | 85 | 46.4 |
| | 101–200 | 43 | 23.5 |
| | >200 | 55 | 30.1 |
| Current CD4+ T counts (cells/μL) | ≤100 | 24 | 13.1 |
| | 101–200 | 19 | 10.4 |
| | >200 | 140 | 76.5 |
| HIV RNA Copies/ mL | ≤1000 | 134 | 73.2 |
| | >1000 | 49 | 26.8 |

## Prevalence of *Cryptococcal Antigenemia*

The overall prevalence of *Cryptococcal Antigenemia* was 7.7% (14/183) of which 71% (10/14) were males (**Table 3**). Among CrAg positive patients, about 86% (12/14) had viral load of >1000 copies/mL and about 71% (10/14) had baseline CD4+ T cell count <100 cells/μL (**Fig 1**).

## Assessment of associated risk factors for *cryptococcosis*

In binary logistic regression, age, marital status, education, occupation, income, residence, distance from a health facility, duration of living with HIV, duration with first-line ART & second-line ART and baseline CD4 count were not significantly associated with *cryptococcal antigenemia* (P > 0.05). Gender, patients' adherence to ART, occupational exposures to the soil, having separated house from domestic animals like hens, self-reported headache & weight loss, current CD4+ T cells, and viral load were associated with a higher prevalence of *cryptococcal antigenemia* in the crude analysis (**Table 4**).

After adjusting for confounding factors, *cryptococcal antigenemia* was significantly higher among males [AOR,95% CI: 4.8(1.14, 20), P <0.05], occupational exposure to soil [AOR, 95% CI: 6.8(1.38, 33), P < 0.05], non-separated house from domestic animals like hens [AOR, 95% CI: 0.06(0.01, 0.51), P < 0.05], poor adherence to ART [AOR, 95% CI: 0.12(0.03, 0.4), P< 0.05], CD4+ T cell counts ≤ 100 cell/μL [AOR, 95% CI: 6.57(1.9, 23.3), P < 0.05] and higher viral load (HIV RNA >1000 copies/mL) [AOR, 95% CI: 11.7(2.4, 57.8), P < 0.05]. Therefore, male HIV/AIDS patients, those occupationally exposed to soil, having house non-separated from domestic animals, poor adherence, low CD4+ T cell count and with high viral load were 4.8, 6.8, 0.06, 0.12, 6.57 and 11.7 times more likely to harbor *cryptococcal antigenemia* than their counters, respectively.

## Discussion

In this study, the prevalence of *Cryptococcosis* among HIV/AIDS patients on second-line ART attending two Western Oromia hospitals was determined. The prevalence in the current study (7.7%) is comparable to a study conducted in Addis Ababa (8.4%) [13] and Indonesia (7.1%) [14]. But this result is higher than a study conducted in North West Ethiopia (0.5%) [15], South West London (5%) [16], Southern Nigeria (5.1%) [17] and Haiti (1.1%) [18]. However, it lower than the reported prevalence of 10.2% from South East Ethiopia [9], 33% from Kenya [19] and 8.9% from Nigeria [20]. These difference might be due to differences in the study population as the current study enrolled only HIV/AIDS patients under second-line ART, the geographical and weather differences as the transmission of *Cryptococcus* is geographically different, differences in laboratory protocols employed and differences in health care system outside Ethiopia.

Age, baseline CD4 count, duration of HIV infection and duration of ART didn't significantly affect the prevalence of *Cryptococcosis* among HIV-infected patients on Second Line ART which is partly in agreement with a previous study and A higher rate of a headache in this study is comparable to other reports in Ethiopia [9]. Unlike previous studies [17, 19, 20], fever, headache and elder age were not significantly associated with an increased risk of *Cryptococcal Antigenemia* in the current study. This possible reason for these differences could be due to better management of the disease in elders, and better prevention of other diseases causing headache and fever.

The high positivity rate of *Cryptococcus* infection in this study implies the burden of *Cryptococcal* infection among HIV/AIDS patients on second-line ART is of public health importance. In this study, the distribution of CrAg among female and male HIV patients was different.

 

**Table 3. Cross-tabulation of CrAg among HIV-infected patients on 2nd line ART at Ambo General Hospital and Nekemte Referral hospitals, Western Oromia, Ethiopia from September 1 to November 30, 2017.**

| Variables | | CrAg LF Test Result | | Total |
|---|---|---|---|---|
| | | Positive | Negative | |
| Age (years) | 18–28 | 0(0%) | 52(100) | 52(100%) |
| | 29–38 | 5(8%) | 59(92%) | 64(100%) |
| | 39–48 | 7(15.2) | 39(84.8) | 46(100%) |
| | 49–58 | 2(12.5) | 14(87.5) | 16(100%) |
| | 59–68 | 0(0%) | 3(100%) | 3(100%) |
| | >68 | 0(0%) | 2(100%) | 2(100%) |
| Sex | Female | 4(3.5%) | 111(96.5) | 115(100%) |
| | Male | 10(14.7) | 58(58.3) | 68(100%) |
| Marital Status | Single | 3(9.1%) | 30(90.9%) | 33(100%) |
| | Married | 6(6.2%) | 91(93.8%) | 97(100%) |
| | Divorced | 3(15.8%) | 16(84.2%) | 19(100%) |
| | Widowed | 2(6.9%) | 27(93.1%) | 29(100%) |
| | Separated | 0(0%) | 7(100%) | 7(100%) |
| Educational Status | Illiterate | 2(3.9%) | 49(96.1%) | 51(100%) |
| | Read & write | 0(0%) | 12(100%) | 12(100%) |
| | Primary School | 9(14.1%) | 55(85.9%) | 64(100%) |
| | Secondary School | 1(2.3%) | 42(97.7%) | 43(100%) |
| | Above college | 2(15.4%) | 11(84.6%) | 13(100%) |
| Occupation | Employee | 4(13.8%) | 25(86.2%) | 29(100%) |
| | Farmers | 4(10.8%) | 33(89.2%) | 37(100%) |
| | Merchants | 1(2%) | 48(98%) | 49(100%) |
| | Students | 0(0%) | 21(100%) | 21(100%) |
| | Housewife | 0(0%) | 22(100%) | 22(100%) |
| | Daily Laborer | 4(30.8%) | 9(69.2%) | 13(100%) |
| | Others | 1(8.3%) | 11(91.7%) | 12(100%) |
| Income | ≤1500 | 12(8.1%) | 137(91.9%) | 149(100%) |
| | >1500 | 2(5.9%) | 32(94.1%) | 34(100%) |
| Residence | Rural | 4(7.8%) | 47(92.2%) | 51(100%) |
| | Urban | 10(7.6%) | 122(92.4) | 132(100%) |
| Years since diagnosed positive for HIV | <0.5 | 2(14.3%) | 12(85.7%) | 14(100%) |
| | 0.5–5.5 | 12(7.1%) | 157(92.9%) | 169(100%) |
| Years on first line ART | <0.5 | 0(0%) | 13(100%) | 13(100%) |
| | 0.5–5.5 | 10(9.3%) | 97(90.7%) | 107(100%) |
| | ≥5.5 | 4(6.3%) | 59(93.7%) | 63(100%) |
| Years on second line ART | <0.5 | 2(9.5%) | 19(90.5%) | 21(100%) |
| | 0.5–5.5 | 10(8.1%) | 114(91.9%) | 124(100%) |
| | ≥5.5 | 2(5.3%) | 36(94.7%) | 38(100%) |
| Adherence | Poor | 10(25.6%) | 29(74.4%) | 39(100%) |
| | Good | 4(2.8%) | 140(97.2%) | 144(100%) |
| The distance of the hospital | <1 Km | 0(0%) | 29(100%) | 29(100%) |
| | 1–3 Km | 7(9.3%) | 68(90.7%) | 75(100%) |
| | >3 Km | 7(8.9%) | 72(91.1%) | 79(100%) |
| Expose to blood | Yes | 1(25%) | 3(75%) | 4(100%) |
| | No | 13(7.3%) | 166(92.7%) | 179(100%) |

(*Continued*)

**Table 3.** (Continued)

| Variables | | CrAg LF Test Result | | Total |
|---|---|---|---|---|
| | | **Positive** | **Negative** | |
| Expose to soil | Yes | 12(14.1%) | 73(85.9%) | 85(100%) |
| | No | 2(2%) | 96(98%) | 98(100%) |
| House separated from the hen | No | 7(43.8%) | 9(56.2%) | 16(100%) |
| | Yes | 2(3%) | 65(97%) | 67(100%) |
| Patient was | Inpatient | 1(12.5%) | 7(87.5%) | 8(100%) |
| | Outpatient | 13(7.4%) | 162(92.6%) | 175(100%) |
| Fever | Present | 3(6.7%) | 42(93.3%) | 45(100%) |
| | Absent | 11(8%) | 127(92%) | 138(100%) |
| Headache | Present | 12(12.9%) | 81(87.1%) | 93(100%) |
| | Absent | 2(2.2%) | 88(97.8%) | 90(100%) |
| Skin Lesion | Present | 5(15.6%) | 27(84.4%) | 32(100%) |
| | Absent | 9(6%) | 142(94%) | 151(100%) |
| Cough | Present | 4(9.3%) | 39(90.7%) | 43(100%) |
| | Absent | 10(7.1%) | 130(92.9%) | 140(100%) |
| Neck Stiffness | Present | 2(25%) | 6(75%) | 8(100%) |
| | Absent | 12(6.9%) | 163(93.1%) | 175(100%) |
| Weight Loss | Present | 6(15.8%) | 32(84.2%) | 38(100%) |
| | Absent | 8(5.5%) | 137(94.5%) | 145(100%) |
| Night Sweat | Present | 4(11.4%) | 31(88.6%) | 35(100%) |
| | Absent | 10(6.8%) | 138(93.2%) | 148(100%) |
| Blurred Vision | Present | 0(0%) | 12(100%) | 12(100%) |
| | Absent | 14(8.2%) | 157(91.8%) | 171(100%) |
| Baseline CD4+ T cell count (Cells/μL) | ≤100 | 10(11.8%) | 75(88.2%) | 85(100%) |
| | 101–200 | 2(4.7%) | 41(95.3%) | 43(100%) |
| | >200 | 2(3.6%) | 53(96.4%) | 55(100%) |
| Current CD4+ T cell count (Cells/μL) | ≤100 | 7(29.2%) | 17(70.8%) | 24(100%) |
| | 101–200 | 1(5.3%) | 18(94.7%) | 19(100%) |
| | >200 | 6(4.3%) | 134(95.7%) | 140(100%) |
| HIV RNA Copies/mL | ≤1000 | 2(1.5%) | 132(98.5%) | 134(100%) |
| | >1000 | 12(24.5%) | 37(75.5%) | 49(100%) |

Male patients had higher seropositivity (71.4%) compared to their female (28.6%) counterparts. In the United States, more men compared to women have been reported to carry a higher burden of cryptococcal infections [21]. The multivariate analysis revealed that there was a significantly higher prevalence of *Cryptococcus* infections in males than in females. This might be due to the fact that males have more outdoor activities in Ethiopia which could expose them to the droppings of avians and which in turn leads to Cryptococcal infection. In contrast to our findings, a study done in Nigeria [22] found out that female patients had a higher prevalence compared to male patients. This difference might be due to working habit and cultural differences between the two countries as males are mostly responsible for outdoor activities in Ethiopia.

The majority of the *Cryptococcal Antigenemia* positive cases (85.7%) were in the age group of 29–38 and 39–48 years in the current study; although, age was not a significant predictor of cryptococcosis. This finding is comparable to the study done in Nigeria [22]. Cryptococcal antigenemia was slightly higher in those with a shorter duration of second-line ART which means second-line ART is playing a critical role in preventing opportunistic infections.

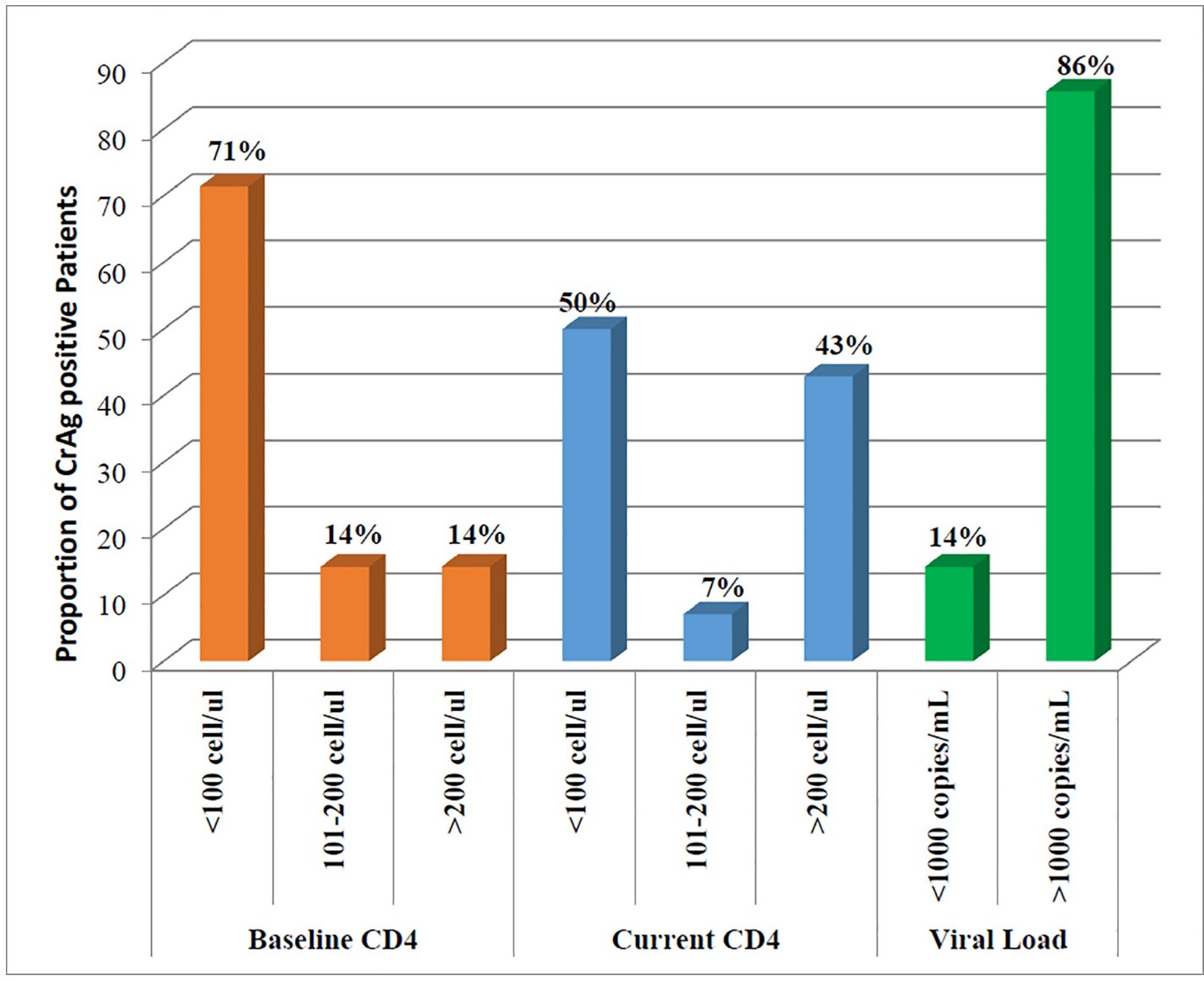

**Fig 1. Baseline and current CD4+ T cell count, and viral load of CrAg positive HIV/AIDS patients on 2^nd line ART at two western Oromia hospitals, Ethiopia.**

The distribution of *Cryptococcal Antigenemia* among patients in this study was highly varying based on CD4+T cell counts as 50% of CrAg positive patients had CD4+ T cell counts below 100 cells/μL. Several studies done in different countries have also reported a consistently higher prevalence of *Cryptococcal Antigenemia* in patients with lower CD4+ T cell counts [23]. Our findings are comparable to other studies done in Uganda [24] and Cambodia [25] where authors reported a higher prevalence of cryptococcal antigenemia associated with low CD4+ T cell counts. The association between the two variables could be due to the fact that low CD4+ T cell counts predispose HIV-infected patients to opportunistic infections because of the immunosuppression. In resource-limited settings, screening of patients with a CD4+ T cell count less than 100 cells/μL for *Cryptococcus* infections may be more clinically relevant as CD4+ T cell tests are currently more accessible.

**Table 4. Assessment of risk factors associated with *CrAg* among HIV patients on second line ART at two hospitals, Western Oromia, Ethiopia from September 1, 2017, to November 30, 2017.**

| Variables | | CrAg LF Test Result | | COR (95% CI) | P-value | AOR (95% CI) | P-value |
|---|---|---|---|---|---|---|---|
| | | Positive N (%) | Negative N (%) | | | | |
| Sex | Female | 4(3.5%) | 111(96.5) | 1 | | 1 | |
| | Male | 10(14.7) | 58(58.3) | 4.8(1.4–15.9) | 0.011* | 4.8(1.14–20) | 0.032* |
| Adherence to ART | Poor | 10(25.6%) | 29(74.4%) | 0.08(0.02–0.3) | 0.000* | 0.12(0.03–0.4) | 0.001* |
| | Good | 4(2.8%) | 140(97.2%) | 1 | | 1 | |
| Exposure to soil | Yes | 12(14.1%) | 73(85.9%) | 7.89(1.71–36) | 0.008* | 6.8(1.38–33) | 0.018* |
| | No | 2(2%) | 96(98%) | 1 | | 1 | |
| Separated house from hen | No | 7(43.8%) | 9(56.2%) | 0.04(0.01–0.2) | 0.000* | 0.06(0.01–0.5) | 0.01* |
| | Yes | 2(3%) | 65(97%) | 1 | | 1 | |
| Headache | Present | 12(12.9%) | 81(87.1%) | 1 | | 1 | |
| | Absent | 2(2.2%) | 88(97.8%) | 0.15(0.03–0.7) | 0.016* | 2.4(0.26–22.8) | 0.4 |
| Weight Loss | Present | 6(15.8%) | 32(84.2%) | 1 | | 1 | |
| | Absent | 8(5.5%) | 137(94.5%) | 0.31(0.1–0.9) | 0.042* | 0.2(0.03–1.13) | 0.06 |
| Current CD4+ T cell count (Cells/μL) | ≤100 | 7(29.2%) | 17(70.8%) | 9.2(2.76–30.6) | 0.000* | 6.57(1.9–23.3) | 0.004* |
| | 101–200 | 1(5.3%) | 18(94.7%) | 1.2(0.14–10.9) | 0.84 | 0.8(0.09–7.6) | 0.85 |
| | >200 | 6(4.3%) | 134(95.7%) | 1 | | 1 | |
| HIV RNA Copies/ml | ≤1000 | 2(1.5%) | 132(98.5%) | 21.4(4.6–99.9) | 0.000* | 11.7(2.4–57.8) | 0.002* |
| | >1000 | 12(24.5%) | 37(75.5%) | 1 | | 1 | |

*significance at P ≤ 0.05, AOR = adjusted odds ratio, COR = crude odds ratio

The prevalence of *Cryptococcal Antigenemia* was 14.28% in patients with a baseline CD4+T cell count of between 101–200 cells/μL. A recent study in Ethiopia reported an estimated serum *Cryptococcal Antigenemia* prevalence of 5.8% among HIV-infected adults with CD4+ T cell counts between 201–350 cells/μL [9]. Unlike this, the current study reported a 42.9% prevalence Cryptococcal *Antigenemia* in patients with current CD4+ T cell counts of >200 cells/μL but this higher prevalence of CrAg among patients with low CD4+ T cell counts is consistent with a study done in South Africa [26]. Differences in culture, lifestyle, using farming as a source of income and other related factors play a significant role for the observed differences across studies as the transmission of *Cryptococcus neoformans* is governed by all these factors.

In this study, high viral load was an independent predictor of *cryptococcal antigenemia* which is in line with a study from Kenya [27]. In general, being male, occupational exposures to the soil, having a non-separated house from domestic animals like hens and chicken, CD4+ T cell counts ≤100 cells/μL and viral load >1000 RNA copies/mL were independent predictors of *Cryptococcal Antigenemia*.

Although this study has limitations like small sample size, non-random sampling technique, and limited study area, it could play an important role in combating death secondary to cerebral meningitis due to *Cryptococcal antigenemia* among HIV/AIDS patients.

## Conclusion

The prevalence of *Cryptococcal Antigenemia* in patients under second line ART is significantly high in the study area. Being male, poor adherence to ART, having a house not separated from domestic animals like hens, occupational exposures to contaminated soil with avian droppings, low CD4+ T cells count and high viral load was significant risk factors that increased *cryptococcal antigenemia*. Public health measures, early diagnosis, and treatment are vital to prevent

*Cryptococcal meningitis* and death among HIV/AIDS patients. Further large-scale longitudinal studies are recommended to explore the risk factors and maximize the benefits of ART.

## Supporting information

**S1 Supporting information. Questionnaires used to collect data from HIV/AIDS patients.** (DOCX)

## Acknowledgments

We would like to acknowledge Wollega University for administrative support, all study participants for their cooperation, and administrative and laboratory staffs of respective hospitals for all their support given to carry out this study. Moreover, we would like to thank the data collectors and study participants without whom the research would not be a reality. We duly acknowledge our families for their patience and motivation.

## Author Contributions

**Conceptualization:** Nuguse Geda, Tafese Beyene, Regea Dabsu, Hylemariam Mihiretie Mengist.

**Data curation:** Nuguse Geda, Regea Dabsu.

**Formal analysis:** Nuguse Geda, Tafese Beyene, Regea Dabsu.

**Investigation:** Nuguse Geda, Regea Dabsu, Hylemariam Mihiretie Mengist.

**Methodology:** Nuguse Geda, Regea Dabsu, Hylemariam Mihiretie Mengist.

**Project administration:** Nuguse Geda, Regea Dabsu, Hylemariam Mihiretie Mengist.

**Resources:** Nuguse Geda, Tafese Beyene, Regea Dabsu.

**Software:** Nuguse Geda, Tafese Beyene.

**Supervision:** Tafese Beyene, Regea Dabsu, Hylemariam Mihiretie Mengist.

**Validation:** Nuguse Geda.

**Writing – original draft:** Nuguse Geda, Regea Dabsu, Hylemariam Mihiretie Mengist.

**Writing – review & editing:** Nuguse Geda, Hylemariam Mihiretie Mengist.

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
