## [Decision Letter · Decision Letter 0]

27 Sep 2019

PONE-D-19-20216

Prevalence of Cryptococcal Antigenemia and Associated Factors Among HIV/AIDS Patients on Second-Line Antiretroviral Therapy at Two Hospitals in Western Oromia, Ethiopia

PLOS ONE

Dear Mr Mengist,

Thank you for submitting your manuscript to PLOS ONE. After careful consideration, we feel that it has merit but does not fully meet PLOS ONE’s publication criteria as it currently stands. Therefore, we invite you to submit a revised version of the manuscript that addresses the points raised during the review process.

REQUIRED: Clarify the CD4 count and the population being studied, as described in the reviewer comments.

REQUIRED: Address All reviewer comments individually in the "Response to Reviewers" document included with the resubmission.

We would appreciate receiving your revised manuscript by Nov 11 2019 11:59PM. To enhance the reproducibility of your results, we recommend that if applicable you deposit your laboratory protocols in protocols.io, where a protocol can be assigned its own identifier (DOI) such that it can be cited independently in the future. For instructions see: http://journals.plos.org/plosone/s/submission-guidelines#loc-laboratory-protocols

We look forward to receiving your revised manuscript.

Kind regards,

Kirsten Nielsen, Ph.D

Academic Editor

PLOS ONE

Journal Requirements:

Additional Editor Comments (if provided):

Reviewers' comments:

Reviewer's Responses to Questions

**Comments to the Author**

1. Is the manuscript technically sound, and do the data support the conclusions?

Reviewer #1: Partly

2. Has the statistical analysis been performed appropriately and rigorously? 

Reviewer #1: Yes

3. Have the authors made all data underlying the findings in their manuscript fully available?

Reviewer #1: Yes

4. Is the manuscript presented in an intelligible fashion and written in standard English?

Reviewer #1: Yes

5. Review Comments to the Author

Reviewer #1: Authorship order is different on the manuscript than in the PLoS One system.

Abstract Results: would prefer you to specify the median age (with interquartile range). % married is of no value and is distracting. Would remove that.

% living with HIV and/or on ART -consider changing units to years instead of months. This is more intuitive/understandable for readers

Introduction:

Line 59 -Define MLICs

Line 61 -CrAg isn't antibodies, it's antigen. would remove the term anti-cryptococcal antibodies. replace with "antigen", same with line 63

Methods:

Line 90 -don't use contractions in formal writing. Same with line 132.

Results:

Line 162: don't capitalize "second line ART"

Line 162 -saying 35% were in one age group is unhelpful. State the median age and interquartile range.

Line 168 -change to years

Line 170 -is this an error? the baseline CD4 was 163 +/- 182? That doesn't make sense.

Line 173 -the prevalence is 7.7%. is this among peoplew ith CD4<100? On line 170 it says mean CD4 was 434. Very confused about the population you're describing.

Line 221: Men tend to have more advanced HIV disease (ie. lower CD4 counts)

Line 229: It's not clear from your results what the CD4 counts were of your patient population and those who were CrAg+

Line 257: higher than what?

In your figure, label your Y axis. Don't use decimal places unless it's useful. And put percentages behind your numbers.

6. PLOS authors have the option to publish the peer review history of their article (what does this mean?). If published, this will include your full peer review and any attached files.

Reviewer #1: No

---

## [Author Response · Author response to Decision Letter 0]

30 Sep 2019

Response to the reviewer

1. Authorship order is different on the manuscript than in the PLoS One system.

Response: We have communicated with the journal office for another paper and confirmed the authorship order in the manuscript is the final acceptable order. The journal has similar rules which we thought will be applicable for this manuscript, too. It was difficult for the corresponding author to put his name last in the editorial manager.

2. Abstract Results: would prefer you to specify the median age (with interquartile range). % married is of no value and is distracting. Would remove that.

% living with HIV and/or on ART -consider changing units to years instead of months.

Response: The age was described with median and IQR and we removed % marriage. And we changed unit of stay with HIV or ART in years instead of months.

3. Introduction: Line 59 -Define MLICs Line 61 -CrAg isn't antibodies, it's antigen. would remove the term anti-cryptococcal antibodies? replace with "antigen", same with line 63

 Response: We defined MLICs (Middle and low income countries) and HICs (High income countries). We removed the term “anti-cryptoccocal antibodies” and changed the “antibody” miswording to “antigen”.

4. Methods:

Line 90 -don't use contractions in formal writing. Same with line 132.

 Response: We fixed the abovementioned issue.

5. Results: Line 162: don't capitalize "second line ART"

Line 162 -saying 35% were in one age group is unhelpful. State the median age and interquartile range.

 Response: We modified the phrase “Second Line ART” into “second line ART” and the median age was used than mean age.

6. Line 168 -change to years. Line 170 -is this an error? the baseline CD4 was 163 +/- 182? That doesn't make sense. Line 173 -the prevalence is 7.7%. is this among people with CD4<100? On line 170 it says mean CD4 was 434. Very confused about the population you're describing.

Response: We changed months to years in line 168. Line 170…it was error. Baseline CD4+ T cell count was 163 +/- 82, not 163 +/- 182. In line 173….the prevalence of CrAg test positivity was 7.7% from all study participants. But 50% prevalence was among participants whose current mean CD4+ T cell count <100 cells/µL. plus the phrase saying “….CD4 was 434….” is about the current mean CD4 T cell count. Thus our study population are those with current mean CD4 T cell count of 434, but their baseline (before starting treatment) was 163+/-82.

7. Line 221: Men tend to have more advanced HIV disease (ie. lower CD4 counts). Line 229: It's not clear from your results what the CD4 counts were of your patient population and those who were CrAg+

Response: Line 221…it doesn’t necessarily mean men has advanced disease than women rather men usually have outdoor activities in Ethiopia which exposes them for Cryptococcosis infection which in turn resulted higher prevalence of CrAg test positivity among men. Line 229……We included the proportion of CrAg positive with respect to CD+ T cell count in results section. 

8. Line 257: higher than what?

Response: Line 260….higher. it is corrected into “….was significantly ‘high’ in the study area”.

9. In your figure, label your Y axis. Don't use decimal places unless it's useful. And put percentages behind your numbers.

Response: The figure was modified accordingly.

---

## [Editor Report · Decision Letter 1]

12 Nov 2019

Prevalence of Cryptococcal Antigenemia and Associated Factors Among HIV/AIDS Patients on Second-Line Antiretroviral Therapy at Two Hospitals in Western Oromia, Ethiopia

PONE-D-19-20216R1

Dear Dr. Mengist,

We are pleased to inform you that your manuscript has been judged scientifically suitable for publication and will be formally accepted for publication once it complies with all outstanding technical requirements.

With kind regards,

Kirsten Nielsen, Ph.D

Academic Editor

PLOS ONE
---

## [Editor Report · Acceptance letter]

22 Nov 2019

PONE-D-19-20216R1 

Prevalence of Cryptococcal Antigenemia and Associated Factors Among HIV/AIDS Patients on Second-Line Antiretroviral Therapy at Two Hospitals in Western Oromia, Ethiopia 

Dear Dr. Mengist:

I am pleased to inform you that your manuscript has been deemed suitable for publication in PLOS ONE. Congratulations! Your manuscript is now with our production department. 

With kind regards,

on behalf of

Dr. Kirsten Nielsen 

Academic Editor

PLOS ONE